# Dust outpaces bedrock in nutrient supply to montane forest ecosystems

S.M. Aciego[1,2], C.S. Riebe[2], S.C. Hart[3], M.A. Blakowski[1], C.J. Carey[4], S.M. Aarons[1], N.C. Dove[3], J.K. Botthoff[5], K.W.W. Sims[2] & E.L. Aronson[4]

Dust provides ecosystem-sustaining nutrients to landscapes underlain by intensively weathered soils. Here we show that dust may also be crucial in montane forest ecosystems, dominating nutrient budgets despite continuous replacement of depleted soils with fresh bedrock via erosion. Strontium and neodymium isotopes in modern dust show that Asian sources contribute 18–45% of dust deposition across our Sierra Nevada, California study sites. The remaining dust originates regionally from the nearby Central Valley. Measured dust fluxes are greater than or equal to modern erosional outputs from hillslopes to channels, and account for 10–20% of estimated millennial-average inputs of bedrock P. Our results demonstrate that exogenic dust can drive the evolution of nutrient budgets in montane ecosystems, with implications for predicting forest response to changes in climate and land use.

[1] Department of Earth & Environmental Sciences, University of Michigan, Ann Arbor, Michigan 48109, USA. [2] Department of Geology & Geophysics, University of Wyoming, Laramie, Wyoming 82071, USA. [3] Department of Life & Environmental Sciences and the Sierra Nevada Research Institute, University of California, Merced, California 95343, USA. [4] Department of Plant Pathology and Microbiology, University of California, Riverside, California 92521, USA. [5] Center for Conservation Biology, University of California, Riverside, California 92521, USA. Correspondence and requests for materials should be addressed to S.M.A. (email: aciego@umich.edu).

Nutrients regulate the distribution of life across Earth's surface. Understanding the relative importance of different sources of nutrients, including underlying bedrock[1] and aeolian dust[2], is therefore a fundamental problem in ecology, biogeochemistry and geobiology[3]. Directly quantifying the importance of dust, which is sensitive to changes in climate and land use[4], is particularly crucial for predicting how ecosystems will respond to global warming and land-use intensification.

Although the mass flux of dust to soils is generally thought to be dwarfed by other fluxes, such as the conversion of bedrock to soil[5], dust can be the dominant source of nutrients to ecosystems where intensive chemical weathering and leaching have depleted underlying bedrock of life-sustaining elements, including phosphorus (P), potassium (K), calcium (Ca) and magnesium (Mg)[2,6,7]. Little is known, however, about the role of dust in montane forest ecosystems, where weathered soil is removed from the surface by physical erosion and replaced with fresh minerals from below by the conversion of bedrock to soil. Modelling suggests that the role of dust is minimal at sites with substantial erosion rates[3,6], but this has not been demonstrated empirically because the relative flux of nutrients from dust and bedrock has never been directly quantified in montane ecosystems. To overcome this limitation, we quantified the relative importance of dust and bedrock in ecosystem nutrient supply across a suite of mountain sites by combining a new seasonal record of dust inputs with existing bulk geochemical and erosion rate data at the same location.

Our analysis confirms findings from previous studies that Asian dust sources account for a fraction of the dust flux in the region. Surprisingly, however, we found that P fluxes from dust are greater than or equal to fluxes from bedrock over the timescales of sediment yield measurements in local streams. Dust is playing a major role in governing the nutrient budgets of our sites. Moreover, the central importance of dust appears to have persisted over the longer timescales of soil formation; paleo-dust records spanning the last million years imply that dust emissions and transport from Asia were greater than or equal to modern rates, suggesting that transoceanic dust has been a substantial long-term source of nutrients to mountain ecosystems of the region. The other main dust source is the more proximal Central Valley, where projected impacts of climate change on drought are likely to enhance dust emissions in the future. This implies that the importance of dust for the ecosystems of the Sierra Nevada may grow in the future, potentially moderating ecosystem adaptation to climate change and land-use intensification. Furthermore, we suggest that the conditions that lead to a large fractional contribution of P from dust relative to bedrock may be widespread in other eroding mountain landscapes and therefore this is likely a worldwide phenomenon.

## Results

**Study area and experimental design.** Our study area lies within the Southern Sierra Critical Zone Observatory (SSCZO), on the western slope of the Sierra Nevada in California (Fig. 1). The surrounding region experiences a Mediterranean-type climate, characterized by cool, wet winters and warm, dry summers. During the summer growing season (approximately mid-May through October), little or no precipitation occurs, providing an opportunity to explore how dust fluxes change over time in response to annual drying conditions. Mean annual precipitation increases and mean annual air temperature decreases with elevation across our study area[8], and since 2012 the entire region has been experiencing a prolonged, historic drought that has killed millions of trees[9]. Additional

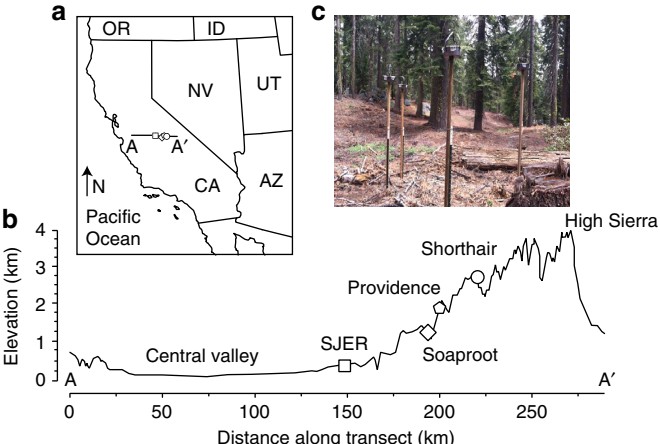

**Figure 1 | Study site locations.** Map (**a**) showing distribution of study sites along an elevation transect (**b**) through the Sierra Nevada, California. Line A–A′ in **a** corresponds to the transect in **b**. The lowest SJER site differs by ~70 km in distance and 2,300 m in elevation from highest Shorthair site in its separation from regional, Central Valley dust sources. Image (**c**) shows the array of dust collectors at the Providence site.

details about these sites can be found in the Methods and previous work[8,10].

To quantify dust fluxes, we deployed passive dust collectors[11] at four sites along a transect spanning ~2,300 m in elevation through the study region (Fig. 1). Dust samples were collected every 1–2 months from June through September 2014 (Table 1), a time period in which the region received no precipitation. To estimate average dust supply rates for each sample, we normalized measured dust masses by collector size (0.048 m$^2$) and deployment time (29–31 days). We dissolved the samples and reserved a set of aliquots for inductively coupled plasma mass spectrometry to measure macro- and micro-nutrient, and trace-element concentrations. On a second set of aliquots, we used ion-exchange chromatography to isolate strontium (Sr), neodymium (Nd) and hafnium (Hf) for analysis of radiogenic isotopes by mass spectrometry. Detailed descriptions of the sampling protocols, and analytical techniques are available in the Methods section.

Extrapolated dust supply rates span an order of magnitude, from 3 to $36 \, \mathrm{g \, m^{-2} \, a^{-1}}$, in agreement with modelled modern dust inputs to the southern Sierra Nevada ($1$–$10 \, \mathrm{g \, m^{-2} \, a^{-1}}$)[12]. Although variable, the dust supply rates estimated here are uncorrelated with both sampling time and location along the elevation transect. For example, the two highest dust supply rates occurred at different sites (Shorthair and Soaproot; see Fig. 1, Supplementary Dataset 1), as well as at different times since the last rainfall event in the Central Valley (August and June, respectively).

The lack of correlation across space and time in dust supply rates contrasts with the clear systematic spatial and temporal trends in radiogenic Sr isotope signatures across our sites (Figs 2 and 3a,c). Dust collected at the lowest elevation site (San Joaquin Experimental Range (SJER)) had a relatively constant $^{87}Sr/^{86}Sr$ over time, while both of the mid-elevation sites (Soaproot and Providence) had progressively lower $^{87}Sr/^{86}Sr$ (that is, a less radiogenic Sr ratio) with increasing time since the last rainfall event. In addition, the average Sr isotope signature of dust was increasingly radiogenic with increasing altitude across our sites (Fig. 3c).

**Dust provenance.** The measured spatiotemporal differences in isotopic signatures likely reflect systematic variations in dust

**Table 1 | Study site locations and dust sampling dates.**

| Site | IGSN | Latitude (°N) | Longitude (°W) | Elevation (m) | Dust sampling dates |
|------|------|---------------|----------------|---------------|---------------------|
| SJER | IE1170005 | 37.10794 | 119.73198 | 400 | 7/5/14; 8/6/14; 9/6/14 |
| Soaproot | IE1170007 | 37.03053 | 119.25817 | 1,100 | 7/6/14; 8/6/14; 9/6/14 |
| Providence | IE1170001 | 37.06008 | 119.18268 | 2,000 | 7/6/14; 8/6/14; 9/6/14 |
| Shorthair | IE1170003 | 37.06698 | 118.98711 | 2,700 | 8/7/14 |

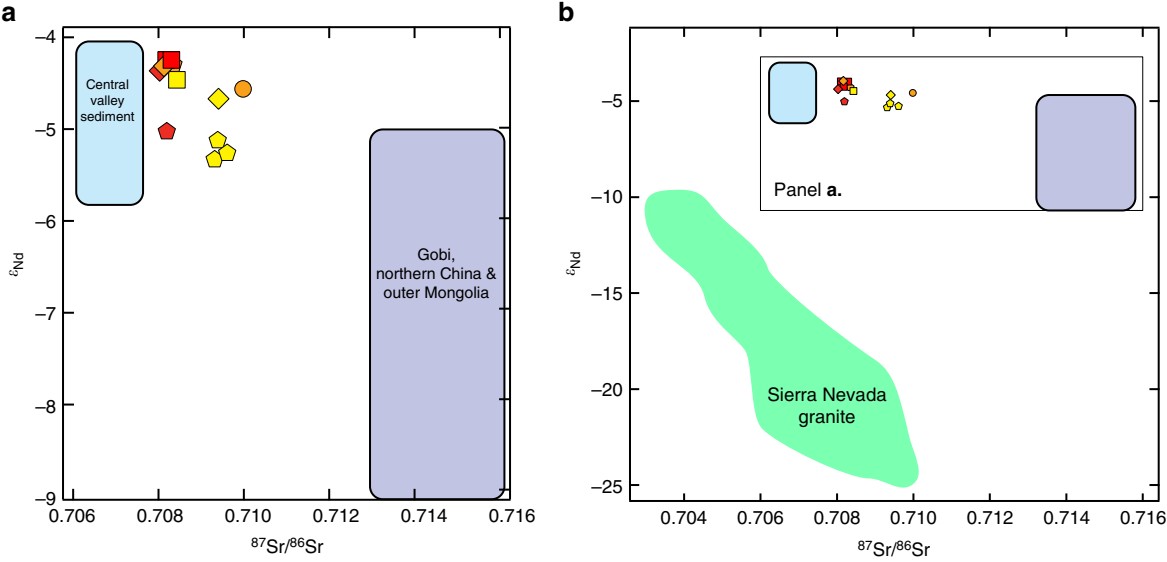

**Figure 2 | Strontium and neodymium dual isotope plot.** Isotopic compositions of Sierra Nevada dust samples are plotted in context of literature derived potential source area isotopic compositions (**a**) Central Valley and Asia (**b**) including Sierra Nevada granite[25,29,32]. External reproducibility (2 s.d.) of new data are smaller than the symbol size; errors for literature data are 0.5–1.0 $\varepsilon_{Nd}$. Symbols for the four sites match those in Fig. 1: circles are Shorthair (elevation ~2,700 m); pentagons are Providence (elevation ~2,000 m); diamonds are Soaproot (elevation ~1,100 m); and squares are San Joaquin Experimental Range (elevation ~400 m). Colours reflect passage of time during the sampling interval: yellow (July), orange (August) and red (September). The relatively radiogenic composition and small range in $\varepsilon_{Nd}$ indicates that local, Sierra Nevada granite provides negligible dust emission and deposition to these sites (**b**); dust is a combination of regional Central Valley sources and global Asian sources (**a**).

contributions from different sources, which can have distinctly different Sr, Nd and Hf isotopic signatures due to differences in bulk geochemistry and geologic age. Sr and Nd are considered robust estimators of source because the radiogenic isotopic compositions should not change during transport, and also show variability between source regions, generally based on the age of the crustal material. Nd isotopes (expressed as the ratio $^{143}Nd/^{144}Nd$) have been used widely to resolve sources areas with very similar Sr isotope ($^{87}Sr/^{86}Sr$) ratios, and the combination of Sr and Nd ratios has been found to be an excellent fingerprint in many cases[13].

Applying isotopic signatures as direct source fingerprints can be problematic because the composition of the transportable source area dust may be difficult to determine. Studies have shown that Sr isotopic ratios are size-fraction dependent, both in aerosols[14] and Asian potential source area samples[15]. Nd isotopes are insensitive to size fraction in loess and desert sands[15–17], although size-dependent variations have been found in some studies[18]. The constraints imposed by long-distance transport thus require the comparison of consistent size fractions between source and deposited dust for global dust sources. Weathering is also known to affect Sr isotopic ratios[19], whereas Nd isotopes have not been found to be affected by chemical weathering[20]. In this study, we use the combined Sr–Nd compositions of the bulk dust samples collected to determine first, the provenance of

exogenic material deposited in the SSCZO, and second, the relative contributions of sources to the total dust load.

The prevailing winds (W-E) and airmass transport to this site (see Supplementary Figs 1–4) combined with the easterly barrier of the high peaks of the Sierra Nevada mountains implies that nearly all aeolian material incident upon our study sites comes from the west. The most likely regional sources to the west are the Coast Ranges, Central Valley and local granitic bedrock of the Sierra Nevada, California. The most likely global source to the west is Asia.

While there are many arid regions in Asia, a combination of aridity, extensive transportable material, sustained winds and high elevation are required to make a source region dominant. A recent synthesis of new isotopic data and a compilation of a decade of work[21] suggests that Mongolia and China have been the dominant dust sources from Asia for the last decade due to increased dust storm activity and extended desertification. Asian and regional sources differ markedly both in $\varepsilon_{Nd}$ and in $^{87}Sr/^{86}Sr$ ratios (Fig. 2). Globally, Asia accounts for more than one third of modern dust emissions, and ice core records indicate it has been a sustained dust source for the past 100 ka (ref. 22). Westerly winds transport Asian dust >1,000 km across the Pacific Ocean to California[23,24], making Asian dust a potentially important contributor to both the modern and long-term average dust supply rates

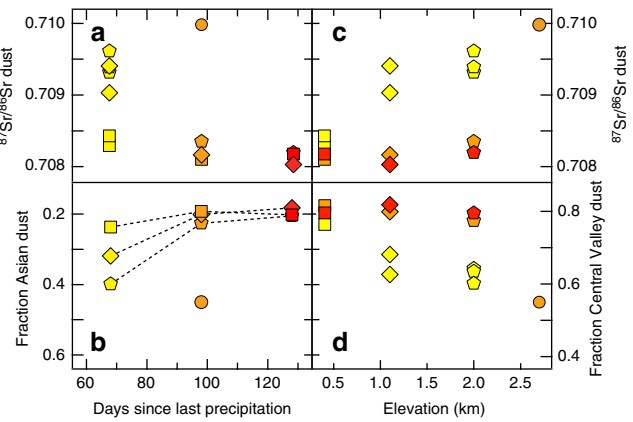

**Figure 3 | Sr compositions and dust contributions.** Seasonal record of strontium (Sr) isotopic composition and relative proportion of Central Valley dust to the total flux incident at the four sites, plotted as a function of time since last precipitation in the Central Valley (data from Friant Dam). Symbols for the four sites match those in Fig. 1: circles are Shorthair (elevation ~2,700 m); pentagons are Providence (elevation ~2,000 m); diamonds are Soaproot (elevation ~1,100 m); and squares are SJER (elevation ~400 m). Colours reflect passage of time during the sampling interval: yellow (July), orange (August) and red (September). (**a**) Sr data from multiple collectors in July indicate the variability in dust composition at a single site (s.d. of the mean of each sample measurement is <0.0015). For sites with multiple months of data, Sr compositions trend towards less radiogenic compositions as the dry season progresses, towards the composition of the Central Valley (Fig. 2). (**b**) Relative proportions of Central Valley versus Asian dust calculated using the Sr compositions and two-component isotope mixing model. The low-elevation site is dominated by Central Valley dust (~80%) throughout the dry season, whereas the higher elevation sites decline in the relative contribution of Asian sources from as much as 40% early in the sampling interval to 20% at the end. (**c**) Sr isotopic composition as a function of elevation, indicating the general trend of more radiogenic compositions with higher elevation. (**d**) Relative proportions of Central Valley versus Asian dust as a function of elevation. The lowest elevation site has the largest fraction of Central Valley dust, while the highest elevation site has the largest fraction of Asian dust.

at our study sites. Extensive work has been done to characterize the Sr and Nd, isotopes of Asian dust sources and Asian dust. Lupker et al.[25] compiled <5 mm size-fraction data for Sr–Nd bulk compositions, that is, the transportable carbonate plus silicate sediment. Based on regional Asian information[21] and a recent data compilation[25], we generalize the Asian dust isotopic composition as falling within the range of dust sources from Mongolia, the Gobi desert and northern China (Fig. 1: $^{87}Sr/^{86}Sr = 0.713$ to 0.716 and $\varepsilon_{Nd} = -5.00$ to $-9.25$).

The influence of regional dust sources may be more limited to the recent (c. 150 years) period of intensive land use and water diversions, particularly in the Central Valley, where the once extensive Tulare Lake and surrounding marshlands were converted into relatively dry and seasonally barren agricultural fields[26,27]. We hypothesize that Central Valley dust sources have become even more important over the last 4 years during the ongoing California drought, due to increases in local dust deflation rates, which contribute to the higher number of poor air-quality days in Central Valley cities[28]. The dominant regional sources of dust to the SSCZO site are constrained by local bedrock and small amount of continental area to the west: the Central Valley and Coast Range of California. Little information is available on the geochemistry—including both bulk chemistry and Sr–Nd isotope—of the Central Valley and

Coast Range potential dust sources, although, there is a small amount of sediment data[29] for river sediments of the San Joaquin River (Fig. 1: $^{87}Sr/^{86}Sr = 0.7060$ to 0.7075, $\varepsilon_{Nd} = -5.8$ to $-4.1$; ranges incorporate external reproducibility). Because the San Joaquin River aggregates sediment from the southern Central Valley, it should provide a good approximation of the composition of material from this region. Rivers draining the more northern portions of the Central Valley have a more radiogenic Sr and less radiogenic Nd composition[29], likely due to the abundance of Neogene volcanics[30]. While there are no other available Nd isotopic measurements for suspended sediments or dust sources on the west side of the Sierra Nevada or the Central Valley, Ingram and Lin[31] found similar $^{87}Sr/^{86}Sr$ compositions for San Joaquin River sediments (0.707 to 0.708) transported to the San Francisco Bay.

While we installed collectors more than two metres above the ground surface, local eddies could drive material from the ground below to the dust collectors via suspension and saltation during short, strong wind bursts. To evaluate this possibility, we also compared the isotopic composition of dust to the Sr–Nd composition of the Sierra Nevada granite bedrock. Extensive work has characterized the range in Sr–Nd compositions[32]. Based on the relatively high $\varepsilon_{Nd}$ compositions of dust ($-5.5$ to $-4.2$) compared to Sierra Nevada granite ($-10$ to $-25$) at the observed range in dust Sr ratios (0.708 to 0.710), we discount the possibility of local granite as a dust source or a source of contamination to the dust collectors (Fig. 2).

Finally, as indicated by the airmass transport modelling (Supplementary Figs 1–4), a fraction of air masses transported to our site travelled over eastern California, Oregon and Washington, and western Nevada in the summer of 2014. The dearth of dust source isotope data from these locations precludes an isotopic comparison of potential dust source to our samples. However, the radiogenic isotopic provinces defined by Farmer and DePaolo[33] indicates that the bedrock source material falls in the range of $^{87}Sr/^{86}Sr = 0.7041$ to 0.7076 and $\varepsilon_{Nd} = -10$ to $-22$, inconsistent with any plausible mixing scenario in Fig. 2.

**Relative contributions of Asian versus Central Valley dust.** The grouping of all the isotopic data along a trend between Asian and Central Valley sources (Fig. 2) implies that the $\varepsilon_{Nd}$ and $^{87}Sr/^{86}Sr$ signatures of the dust can be straightforwardly explained by simple mixing from two sources: one transoceanic and the other regional. We therefore use the end-member isotopic compositions of Asian and Central Valley sources to estimate their fractional contributions to dust supply at each sampling site. While Nd isotopes provide a rationale for discounting certain materials as sources, the small differences in composition between San Joaquin River sediments (Central Valley sources) and Asian dust means that we cannot calculate a relative contribution based on Nd isotopic compositions as Chadwick et al.[2] have done. Instead, because of the large difference in Sr compositions between sources and the fact that the samples are dust rather than soils (that is, weathering will not fractionate Sr isotopes in the dust during the dry collection period), we calculate the relative contributions of Central Valley and Asian dust using a basic two-component mixing method[34]:

$$\begin{aligned} F_{Asia} &= M_{Asia}^{Sr}/(M_{Asia}^{Sr} + M_{CV}^{Sr}) \\ &= (d_{dust} - d_{CV})/(d_{Asia} - d_{CV}) \end{aligned} \tag{1}$$

where $F_{Asia}$ is the mass fraction of strontium from Asian dust of the total, $M$ is the mass of Sr, and $d$ is the $^{87}Sr/^{86}Sr$ of the dust and dust contributions (that is, the subscripts dust denotes

the collected dust, CV denotes Central Valley mass or composition and Asia denotes Asian mass or composition). For this calculation, we use the mean values of the San Joaquin River sediments (0.7067) and the Gobi-northern China-outer Mongolia dust sources (0.714) to represent the Central Valley and Asia, respectively.

Sr concentrations in loess and dust source areas from Asia vary from 116 to 276 $\mu g\,g^{-1}$ (ref. 16), identical to the range in California river sediment[31]. Given that Sr concentrations are the same for Asia and the Central Valley, the mass fraction of Sr is the same as the total mass fraction. Calculated mass fractions of Asian and Central Valley dust are provided in Fig. 3. If the isotopic composition of the endmember for Asia is instead $^{87}Sr/^{86}Sr = 0.727$, representing the mean composition of the Taklamakan Desert and Ordos Plateau instead of the Gobi Desert, then the Asian mass fraction would decrease to 0.07–0.16. Conversely, if the Central Valley composition is less radiogenic (for example, $^{87}Sr/^{86}Sr = 0.705$ due to incorporation of young volcanics), then the Asian mass fraction increases to 0.34–0.55.

At all sites throughout the summer, Asian dust accounted for more than 18% of the total dust supply. Averaged over the summer, the estimated fractional contribution of Asian dust increased with increasing elevation, from 20% at the lowest site to a maximum of 45% at the highest elevation site (Fig. 3d). Dust collected from the lowest elevation site (SJER) had a relatively constant fractional contribution of Asian and Central Valley dust over the sampling period (Fig. 3d). This is reflected in the constant $^{87}Sr/^{86}Sr$ ratio at this site over time (Fig. 3a). In contrast, the mid-elevation sites (Soaproot and Providence) both showed decreasing $^{87}Sr/^{86}Sr$ over time, consistent with a decrease in Asian dust contributions and an increase in Central Valley dust contributions with time since the last rainfall in Central Valley source areas (Fig. 3b).

The overall dust supply rate did not change systematically over the period of observations (see Supplementary Dataset 1). We can therefore infer that the decrease in the relative contribution of Asian dust at the two mid-elevation sites (Fig. 3d) reflects changes in dust supply rates from the two sources. We hypothesize that the implied decrease in supply from Asia reflects a seasonal decline in Asian windstorms[23], while the increase in supply from the Central Valley reflects a seasonal decrease in regional soil moisture[35]. Temporal differences in the sources of dust between the low-elevation site and the mid-elevation sites are likely due to differences in airflow not observed in global modelling (for example, the Hysplit modelling in Supplementary Figs 1–4). While the Shorthair site helps quantify the spatial pattern in dust composition across the elevation gradient, we cannot use it infer a temporal pattern at the highest elevation in our study, because we only sampled it once.

Calculated contributions from global and regional sources are supported by an additional radiogenic isotope system—lutetium (Lu)-Hf. While Lu–Hf and Sm–Nd are closely linked in the Earth's mantle, they vary significantly and systematically in Earth surface materials[36–41]. In particular, mineral sorting during weathering and transport can decouple the $\varepsilon_{Nd}$ and $\varepsilon_{Hf}$ isotope systems because of the high concentration of Hf in relatively dense, chemically resistant zirconium (Zr) bearing minerals. For example, zircon, which is considered the main reservoir for Zr in terrestrial rocks, is also a major reservoir of Hf because of chemical substitution. Therefore, if the concentration of zircon varies across a series of dust samples, the $\varepsilon_{Hf}$ will also vary while the $\varepsilon_{Nd}$ remains approximately constant (known as the 'zircon effect', for example, ref. 39). Zircon contains large amounts of non-radiogenic Hf due to low initial Lu/Hf ratios and the slow radiogenic decay of $^{176}Lu$ to $^{176}Hf$. Thus, it can be

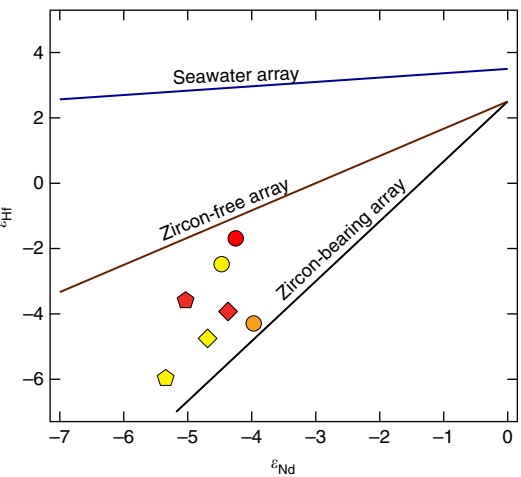

**Figure 4 | Hafnium and neodymium dual isotope plot.** Arrays of Earth's oceans, sediments and igneous rocks (compiled by Bayon and others[40]) are plotted with the hafnium (Hf)-neodymium (Nd) isotope compositions of dust measured in this study. All samples lie within the zircon-free and zircon-bearing array, suggesting that they originate from a combination of near (regional Central Valley) and far (Asian) dust sources. Symbols for the three sites match those in Fig. 1: circles are Shorthair (elevation ~2,700 m); pentagons are Providence (elevation ~2,000 m); diamonds are Soaproot (elevation ~1,100 m); and squares are San Joaquin Experimental Range (elevation ~400 m). Colours reflect passage of time during the sampling interval: yellow (July), orange (August), and red (September).

generalized that the continental budget of Hf is effectively split between a low $\varepsilon_{Hf}$ reservoir in zircon-bearing sediments, and a more radiogenic, zircon-free reservoir[39].

Physical weathering, erosion and transport processes of continental material separates coarser, sandy sediments with abundant zircons from clay-sized (zircon poor) sediments, resulting in more radiogenic $^{176}Hf/^{177}Hf$ (less negative $\varepsilon_{Hf}$) in fine clay and silt-sized sediment. Even more extreme fractionation in the Hf isotope system occurs during dissolution; preferential weathering of the non-zircon portions of the upper continental crust results in the very radiogenic global $\varepsilon_{Hf}$ seawater array[36,41]. Two recent studies[37,41] extended the concept of incongruent continental weathering to dust transport studies. The retention of non-radiogenic Hf in weathering-resistant zircon grains will disproportionately affect the observed $\varepsilon_{Hf}$ relative to $\varepsilon_{Nd}$ in dust particles advected during dust storms. Due to density-driven fallout, the loss of zircon outpaces that of other less-dense minerals during advection through the atmosphere, which results in a systematic increase in the $\varepsilon_{Hf}$ value away from the bulk silicate earth value with travel distance. Modelling of this 'zircon effect' indicates that zircon loss is complete when dust travels >3,000 km from its source[37].

In the context of the zircon fallout effect, our Hf isotope data support the estimated relative contributions of Asian and Central Valley dust. Figure 4 shows the dust sample $\varepsilon_{Hf}$ and $\varepsilon_{Nd}$ data with lines indicating where the data would lie if samples contained concentrations of zircon similar to the bulk silicate earth (zircon-bearing sediment array) or were free of zircon (zircon-free array; defined by Bayon et al.[40]). All of the data lie between the two arrays, suggesting a mixture of dust sources that are close (for example, the Central Valley) and >3,000 km away (for example, Asia). Without endmember compositions of the dust $\varepsilon_{Hf}$ and $\varepsilon_{Nd}$ composition before transport, we cannot use the Hf system to calculate relative

contributions. Nevertheless, these data provide valuable qualitative support to the conclusions drawn from the Sr and Nd results.

**Nutrient supply rates from dust and bedrock**. The relative contributions of Asian and regional sources estimated from our dust measurements (Fig. 3) are consistent with results from previous studies[23,42]. However, here we also compare, for the first time, nutrient fluxes from exogenic and bedrock sources in a landscape where the supply of nutrients from regolith production is significant. This comparison is possible because we can estimate nutrient inputs from bedrock by combining existing measurements of soil production rates together with bulk geochemistry of bedrock for catchments near the Providence dust collection site. Thus, we can capitalize on existing data to quantitatively explore the relative importance of dust and bedrock sources in nutrient fluxes to montane forest ecosystems.

One impact of dust on ecosystems is that exogenic material often contains higher macro- and micro-nutrient concentrations than the bedrock from which dust is derived[43]. The most widely recognized dust-derived limiting nutrient is Fe in the oceans[4] and P in terrestrial ecosystems[44]. For this reason, we focus on P in our comparison of dust and bedrock nutrient concentrations. The average P concentration in dust collected in our study is $1.5 \pm 0.2 \, \mathrm{mg \, g^{-1}}$ (see Supplementary Dataset 1), which is 2.5 times higher than the average bedrock P concentration in granitic plutons of the surrounding region $(0.61 \pm 0.02 \, \mathrm{mg \, g^{-1}})$[45]. The difference is even larger (that is, more than a factor of ten) for the more leucocratic plutons, which have average bedrock P concentrations as low as $0.12 \pm 0.01 \, \mathrm{mg \, g^{-1}}$ (ref. 45).

To estimate supply rates of dust-derived P to the ecosystem, we calculated the overall mass flux from dust to soil and multiplied it by the measured concentration of P in dust. To quantify the overall flux of dust, we extrapolated the mass collected during the sampling interval into an estimated annual flux. This was partitioned into dust fluxes from Asian and Central Valley sources using Sr isotopes according to the mixing equation above. The data presented here represent the summer dry season at these sites; dust deposition in the winter months will occur as both dry fall and wet fall (as precipitation nuclei). As an additional estimate of the annual, total dust flux, we extracted spatially explicit data for the Providence site from a global model of modern dust fluxes[4]. To calculate P supply rates from dust (right half of Fig. 5), we multiplied the measured P concentration in the dust by each of the measured and inferred dust deposition rates (that is, total, Asian, Central Valley and modelled).

To put the estimated P supply from dust in context with supply rates from production of soil from bedrock in the area, we used two types of proxy data: (1) cosmogenic nuclide-based estimates of erosion rates[45,46], which quantify the throughput of minerals in soils over the millennial timescales of soil formation; and (2) sediment trapping[10], which averages erosion rates over the annual to decadal timescales of human observations. Whether they are averaged over short or long timescales, soil production rates provide a proxy for the rate that fresh nutrient-bearing minerals are introduced to the ecosystem from underlying bedrock[6].

Our long-term average soil production rates are based on cosmogenic [10]Be measured in quartz from three types of samples[45]. First, sediment from streams draining forested catchments. Second, samples of exposed bedrock from bare-rock landscapes. Third, samples of sediment from streams draining bare-rock landscapes (see Supplementary Database for individual measurements). The estimated erosion rates of

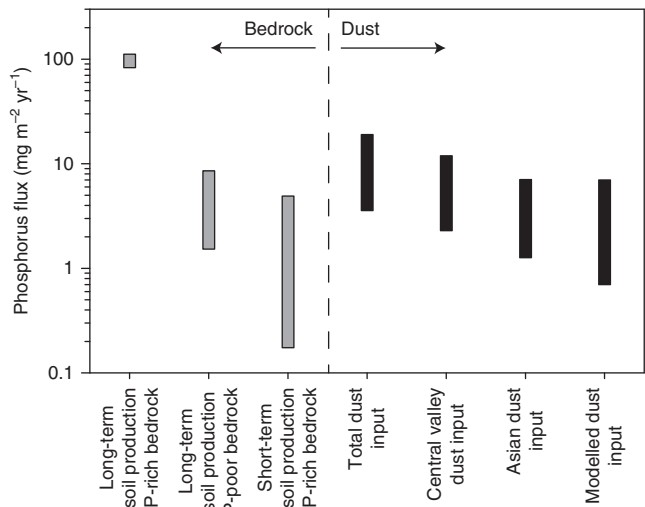

**Figure 5 | Phosphorus fluxes.** Fluxes of the plant-essential macronutrient phosphorus at the Providence Creek study site due to erosion (grey bars) and aeolian deposition (black bars). Bars span ranges in fluxes from multiple measurements (see Supplementary Dataset 2). Total dust flux is the sum of fluxes from Asian and Central Valley sources. On soil-mantled slopes, the total input of P from dust accounts for 10–20% of the supply of bedrock P implied by coupling long-term erosion rates with concentrations of P in bedrock. On bare-rock slopes, the estimated supply of bedrock P is much lower and commensurate with the fluxes from the Asian and Central Valley dust sources, due to slower erosion and lower concentrations of P in bedrock. The fluxes of P implied by catchment-wide sediment yields averaged over a recent seven-year interval are generally slower than the estimated dust fluxes, implying that the modern ecosystem is strongly influenced by the day-to-day contributions of dust from Asia and the Central Valley.

forested catchments of Providence Creek range from 103 to $175 \, \mathrm{g \, m^{-2} \, a^{-1}}$. These are all substantially faster than erosion rates of bare-rock landscapes, which range from 18 to $69 \, \mathrm{g \, m^{-2} \, a^{-1}}$[45]. They are also substantially faster than the 0.2 to $6.1 \, \mathrm{g \, m^{-2} \, a^{-1}}$ average erosion rates inferred from volumes of sediment collected in traps at the forested catchment mouths over the 7-year sampling period from 2003 to 2009 (ref. 10). Because bedrock P concentrations are intrinsic to the landscape, and thus can assumed to be constant over time, the discrepancy in soil production rates over time indicates that the modern flux of bedrock-derived P through the forested ecosystem is much slower than the long-term average flux recorded by the cosmogenic nuclides (Fig. 5).

The difference between the erosion rates of the forested and bare-rock sites likely reflects an inherent difference in the rates of weathering between exposed and soil-mantled bedrock[45]. For example, bare-rock presumably weathers slower in part because it stays drier than soil-mantled bedrock most of the time[47]. Meanwhile the differences in erosion rates across timescales may reflect the fact that the timescales of sediment trapping are too short to capture large, infrequent events that contribute substantially to the long-term averages inferred from cosmogenic nuclides[48].

To use the long-term cosmogenic and short-term sediment trapping data as proxies for soil production rates, we must adopt the simplifying assumption that soils are in a state of approximate dynamic equilibrium across the landscape, with the rate of soil production roughly equal to the rate of soil erosion over the timescales of interest[49]. This should be a reasonable assumption in landscapes where soil thickness and production

rate are coupled in a negative feedback, as may often be the case according to theoretical and empirical studies[50–52].

To estimate supply rates of bedrock-derived P to the ecosystem (left half of Fig. 5), we multiplied the various proxy measurements of soil production rates by the average bedrock P concentrations for each sampling site[3]. To quantify P concentrations in bedrock, we used previously published data on bedrock geochemistry from samples collected near the erosion rate sampling sites[45]. For completeness, we report the bedrock geochemistry data from individual samples in Supplementary Dataset 1. These data were grouped by erosion rate sampling location and the average element concentrations of each group were paired with the corresponding erosion rate for estimates of the supply rates of individual elements from bedrock to soil.

Our analysis shows that the total P flux from dust is greater than or equal to the modern bedrock-derived P flux inferred from the sediment trapping data (Fig. 5). This is true both when we use data collected from our sites, and when we calculate dust-derived P fluxes as the product of modelled dust fluxes[4] and the average P concentration of the continental crust[53]. The measured and modelled dust-derived P fluxes are also both on par with the long-term average bedrock-derived P fluxes for slowly eroding, P-poor, bare-bedrock sites (Fig. 5), where the lack of forest cover may reflect a nutrient limitation that is intrinsic to the underlying bedrock[45]. The modelled dust flux includes inputs from Asian sources, which have likely either remained roughly constant or decreased in magnitude over the last 100 ka (refs 2,4,22). The agreement between the modelled dust fluxes and the inferred long-term average bedrock fluxes at the bare sites therefore implies that exogenous dust can be a vital long-term contributor of nutrients to eroding montane ecosystems with P-poor bedrock. In contrast, the contribution of bedrock-derived P has been considerably higher than the contribution of P from dust over the long term (Fig. 5), because the soil-mantled, forested catchments have both faster millennial-average soil production rates and higher bedrock P concentrations. However, over the short term at these sites, dust is evidently outpacing bedrock in P supply (Fig. 5), suggesting that dust is currently a vital factor in the evolution of ecosystem P pools, even at sites underlain by relatively P-rich bedrock.

**Biogeochemical implications.** The inferred modern contribution of Central Valley dust is probably faster than the long-term average, due to recent drying of Central Valley lakes, agricultural activity and the ongoing, prolonged drought[54]. Future dust sources and transport pathways are uncertain and difficult to assess[55], but if dust fluxes continue to increase as climate warms and land-use intensifies, Sierra Nevada ecosystems will likely develop in response to higher nutrient loading. Atmospheric deposition of anthropogenic P has been invoked to explain recent eutrophication in the nearby Sierra Nevada lakes and also lakes in the Rocky Mountains[56,57] to the east. Forest ecosystems may also be affected by anthropogenic P inputs, albeit more moderately due to their relatively large pools of nutrients. Mature western conifer forests take up P from soil at rates of 310–500 $mg\,m^{-2}\,a^{-1}$ (refs 58–60), about an order of magnitude faster than the modern supply from bedrock weathering and dust deposition, consistent with efficient recycling of P within the ecosystem. If P removal mechanisms and rates remain roughly constant, then increasing atmospheric deposition will enlarge the forest ecosystem P pool over decades to centuries. Thus, our analysis highlights potential contrasts between montane lakes and forests over the timescale of response to climate and land-use changes.

Whereas prior studies have concluded that exogenic dust is an important source of P for ecosystems developed on intensely weathered soils[2,6], here we integrated multiple data sources, including the first direct comparison of P supply from bedrock and dust, to show that the P flux from dust is between 10 and 100% of the total P input to a non-tropical, actively eroding montane ecosystem. Dust deposition rates from our collectors are consistent with modelled rates in mountain ranges around the world[4]. Likewise, erosion rates of both bare and soil-mantled bedrock at our sites are consistent with the global database of erosion rates from granitic mountain landscapes[45]. In addition, the granites of the Sierra Nevada are similar in composition—including P concentration—to many of the granites that underlie 5.7% of the global land surface and form the core of many of the world's mountain ranges[45,61]. We therefore suggest that the conditions that lead to a large fractional contribution of P from dust relative to bedrock may be widespread in eroding mountain landscapes.

Together, our results and analyses indicate that paradigms need to be revised to recognize the potential significance of exogenic dust in the supply of plant-available nutrients to eroding, temperate montane landscapes across the globe.

## Methods

**Sampling dust.** We collected dust using passive collectors consisting of Teflon-coated round pans (25.4 cm in diameter) filled with quartz marbles suspended from the pan bottom with Teflon mesh and capped with an overarching Teflon cross-brace covered in Tanglefoot to discourage roosting by birds. All materials in contact with dust (that is, pan, marbles, and Teflon mesh) were pre-washed before installation in a class 10,000 clean room at the University of Michigan using first bleach, then distilled 2 M HCl, and finally distilled 3 M HNO₃ with rinses of >18.2 MΩ water between each reagent cleaning step. The collectors were deployed on ∼2-m tall wooden posts in open areas at each of the field sites to minimize local contributions of material from wind-suspended soil (that is, from saltation) and nearby trees. Figure 1 shows collectors deployed at the Providence site.

We deployed collectors at SJER, Soaproot and Providence sites on 4 April 2014, and at Shorthair on 11 June 2014. The later deployment date of at the Shorthair site was necessary due to snowpack, which prevented access until June. Sampling dates are reported in Table 1. In July, two dust collectors were sampled from SJER and Soaproot, while three were sampled from Providence. In the subsequent months, one collector was sampled from each site (with the exception of Shorthair, which was sampled only in August). Samples from the other collectors in July, August and September were retained for other analyses. To recover and archive dust samples from each collector on these dates, we rinsed the marbles with >18.2 MΩ cm water into the Teflon-coated pan, removing the marbles and mesh, then transferring the water and dust suspension to pre-cleaned 1 litre LDPE Nalgene bottles. No samples were combined for analyses. Bottles were shipped to the University of Michigan for further processing.

**Sample processing.** The samples were first dried and massed and then prepared for elemental and isotopic analysis in a class 10 laminar flow hood inside a 10,000 level clean room at the University of Michigan, using published procedures[38]. Approximately 10 mg of sediment were weighed and digested in concentrated HF in 3 ml Savillex beakers, sealed within steel-jacketed Parr bombs. Loaded Parr bombs were placed in a 220 °C oven for 48 h. The solutions were carefully dried down on a hot plate, and the procedure was replicated with 6 M HCl at 180 °C for 12–16 h. After the final dry-down, 1 ml of 9 M HCl was added in preparation for column chemistry. Solutions were then split for elemental analysis by inductively coupled mass spectrometry (ICPMS) and ion-exchange chromatography and subsequent radiogenic isotope analysis, using previously published multi-column procedures[62]. All data are presented in Supplementary Dataset 1.

**Elemental analyses.** Trace and major elemental concentrations, including the plant-essential nutrient phosphorus (P), were measured in triplicate on the Thermo Scientific ELEMENT2 ICPMS at the University of Michigan Keck Laboratory in pulse counting mode. The digested dust sample splits were dried and dissolved in 2 ml 2% HNO₃ solutions. An acid blank and reference standards were run every five samples to assess long-term reproducibility and accuracy. Repeat measurements of international standard NIST1640a are provided in previously published work[63]. Baseline detection measurements from the total procedural blank indicate that blanks were never >10% the concentration, even for the lowest concentrations.

**Neodymium radiogenic isotope analysis.** Each sample was first loaded into a 700 μl PFA column filled with 50–100 mesh TruSpec resin. Following procedures published elsewhere[62], we separated Sr and calcium (Ca) from the High Field Strength Elements (HFSE) and the Rare Earth Elements (REE).

We isolated Nd using ion-exchange column chemistry involving two columns. The eluted volume containing both the HFSE and REE was loaded into a 1.5 ml PFA column filled with 100–200 μm LnSpec resin, where the introduction of sequential acids of varying normality resulted in the separation of the REEs from the HFSE, including Hf. The REE cut was subsequently loaded onto a preconditioned 2 ml PFA column filled with clean 50–100 μm LnSpec resin. The subsequent volumes eluted with HCl isolated the Nd from the REE fraction[62]. An acid blank and reference materials of known composition (AGV-2, BCR-2) were processed using the same procedure to ensure long-term reproducibility and to assess error.

Nd was loaded onto outgassed rhenium double filaments after 1 μl of 1 M HCl–1 M HNO$_3$ was added to each dried-down sample. A current of 0.8 A was run through the filament until it was dry. The current was slowly increased to 1.8 A and remained constant for 1 min. The current was then flashed at 2.2 A and decreased to 0 A.

We quantified isotopic ratios using a Thermo-Finnigan Triton Thermal Ionization Mass Spectrometer (TIMS) at the Glaciochemistry and Isotope Geochemistry Laboratory in the Department of Earth and Environmental Science at the University of Michigan following methods outlined elsewhere[37]. Instrumental mass bias was corrected for by applying an exponential mass fractionation law with the $^{146}Nd/^{144}Nd = 0.7219$, and mass 149 was monitored for samarium (Sm) interference. Amplifier gains and baselines were run before each set of analyses. Our measurement of the Nd isotopic standard JNdi-1 (10 ng) was $^{143}Nd/^{144}Nd = 0.512099 \pm 0.000016$ (2 s.d., $n = 8$), which is in agreement with the accepted JNdi-1 standard value of $^{143}Nd/^{144}Nd = 0.512115$ (ref. 64). All data were normalized to the accepted value of JNdi-1. The $^{143}Nd/^{144}Nd$ of AGV-2 and BCR-2 were in agreement with values from the literature[65,66]. Results are presented in epsilon notation, where $\varepsilon_{Nd} = ((^{143}Nd/^{144}Nd_{measured} - {}^{143}Nd/^{144}Nd_{CHUR})/{}^{143}Nd/^{144}Nd_{CHUR}) \times 10,000$.

**Strontium radiogenic isotope analysis.** Splits from the initial TruSpec columns containing the Sr were dried on a hot plate and dissolved in 7.5 M HNO$_3$. Samples were loaded in 500 μl 3 M HNO$_3$ onto Sr columns containing 150 μl Eichrom Strontium specific resin bed in 500 μl 3 M HNO$_3$.

The column was washed and eluted with several stages of HNO$_3$ following the procedure outlined elsewhere[62]. The procedural Sr blank was < ∼60 pg, constituting <0.1% of the total Sr analysed for a typical Sr analysis. Sr samples were loaded onto outgassed 99.98% Re filaments in 1.0 μl concentrated HNO$_3$ along with 0.8 μl TaF$_5$ activator to enhance the ionization efficiency of Sr (ref. 67).

Sr isotope measurements were carried out on a Thermo-Finnigan Triton Plus TIMS using the method outlined previously[68]. For natural runs, fractionation caused by mass-dependent evaporation during machine analysis was corrected for using $^{86}Sr/^{88}Sr = 0.1194$. External precision on the standard runs (NBS987) for $^{87}Sr/^{86}Sr$ was $0.710264 \pm 0.000016$ (2 s.d., $n = 50$), and all values were normalized to the NBS987 accepted value of 0.710245. The $^{87}Sr/^{86}Sr$ of BCR-2 and AGV-2 were in agreement with literature values[69].

**Hafnium radiogenic isotope analysis.** Isolation of Hf was performed through ion-exchange column chemistry from the first step of the Nd isolation as described elsewhere[62]; the column chemistry separates Hf from lutetium (Lu), ytterbium (Yb) and tungsten (W; isotopes with isobaric interferences). Isotopic ratios were measured using a Thermo-Finnigan Neptune Multicollector Inductively Coupled Mass Spectrometer (MC-ICPMS) in the Department of Geology & Geophysics at the University of Wyoming as described elsewhere[70]. Instrumental mass bias was corrected for by applying an exponential mass fractionation law with the $^{179}Hf/^{177}Hf = 0.7325$ and masses 182, 175 and 172 were monitored for W, Lu and Yb interferences. Baselines were measured before each sample, and amplifier gains were measured at the start of every day. Interference corrections to $^{176}Hf/^{177}Hf$ were <100 μg g$^{-1}$. Bracketing measurements of the Hf standard JMC475 were run between every sample, averaging $^{176}Hf/^{177}Hf = 0.282115 \pm 0.000008$ (2 s.d., $n = 83$), which is in agreement with the prior measurements[37,62,70]; all samples were normalized to JMC475 $^{176}Hf/^{177}Hf = 0.282160$. The $^{176}Hf/^{177}Hf$ of BCR-2 and AGV-2 were in agreement with the literature values[66]. Samples are presented in epsilon notation, where $\varepsilon_{Hf} = ((^{176}Hf/^{177}Hf_{measured} - {}^{176}Hf/^{177}Hf_{CHUR})/{}^{176}Hf/^{177}Hf_{CHUR}) \times 10,000$.

**Airmass back trajectory modelling.** Back trajectories using the NOAA HYSPLIT model[71–73] were calculated for each one of the sampling sites (latitude and longitude in Table 1) at an elevation ∼10 m above the surface (Supplementary Figs 1–4). Eighty models were run—5 weeks each month for the summer of 2014 at each of the sample sites. This model calculates the 72 h back trajectory of air masses arriving to the site based on archived meteorological conditions and a local topography using a 0.5 degree grid. The modelling confirms that air masses generally trend from the west and north of the sampling location, with most of the air masses passing over the Central Valley prior to arriving at our sample sites.

But for some of the year air masses also cross over parts of Nevada, Oregon and Washington. Clearly, deflation and incorporation of dust from potential source areas depends on the aridity, availability of material, elevation and windiness, which would need to be assessed before pinpointing a source location on any of the back trajectories.

**Data availability.** The data that support the findings of this study are available from the authors upon request.

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

## Acknowledgements

This work was funded by grants from the David & Lucile Packard Foundation, and the National Science Foundation under grants EPS-1208909, EAR-1331939, EAR-1449197 and ICER-1541047.

## Author contributions

Project planning and design, sample collection and measurement, data analysis, and manuscript preparation by S.M.Ac.; project planning and design, sample collection, data analysis and manuscript preparation by C.S.R.; project planning and design, sample collection, data analysis and manuscript preparation by S.C.H.; sample collection and measurement, data analysis and manuscript preparation by M.A.B.; sample collection and measurement, data analysis and manuscript preparation by C.J.C.; data analysis and manuscript preparation by S.M.Aa.; sample collection and manuscript preparation by N.C.D.; sample collection and manuscript preparation by J.K.B.; sample measurement by K.W.W.S.; project planning and design, sample collection, data analysis and manuscript preparation by E.L.A.

## Additional information

**Competing interests:** The authors declare no competing financial interests.

