## [Peer Review File · Nature Communications]

Reviewers' Comments:

Reviewer #1 (Remarks to the Author)

This manuscript makes use of Sr and Nd isotopes measured in dust collected from a west to east transect into the heart of the Sierra Nevada in California. The isotope data allow the authors to distinguish between local and distance sources of dust, which in this region they argue must be coming from Asia – they assume that the source is most likely the Gobi Desert. They find that between about 20 and 45% of the dust is derived from Asia and that the fraction is greater at the higher elevation sites (further away from local sources in the central valley). They then use the dust flux data in aggregate to calculate the amount of phosphorus (P) deposited on the forests at their higher elevation sites and to compare that with the amount of P delivered to the ecosystem by weathering of bedrock (here they assume that the erosion rate calculated using datasets that provide short-term and long-term erosion rates). They conclude that on sites with little soil and lots of bare rock most of the P comes from dust because of slow rates of soil production and that that fraction declines in locations where a thicker soil cover and hence faster soil production occurs. Finally they argue that given the increase in local dust associated with land-use intensification and potentially with increases in distance dust for the same reason, it is likely that these mountain forests and perhaps others like them elsewhere in the world are likely to see increased nutrient supplies in the future.

This is an excellent paper that combines contributions from isotope geochemists, geomorphologists and ecosystem scientists. The paper is well argued and well supported by the supplemental material. I have a few questions.

The crux of the isotope argument is that there are distinct isotopic signatures between the regional central valley sources and the long distance Asian sources. In the body of the text there is no discussion of other possible regional sources (I am thinking sources derived either from the northeast or the southeast of the mountains). The supplemental material contains a spirited defense of why one does not need to be concerned about those sources and earlier work on soils tends to support the idea that dust contributions from the basin and range drop off as one moves from the White Mountains or the Lahontan Basin into the Sierra Nevada. Still it might be good to bolster the argument with a sentence or two in the main body of the text since many will not interrogate the supplemental material. And for completeness it would be good to provide some sense of the isotope values for basin and range dust in the supplemental material.

The dust data were collected during the summer of 2014. There is no discussion of what the winter delivery of mineral aerosol might be like. I could imagine that the delivery of local/regional dust could be suppressed by wet conditions but what is going to happen to the long-distance material? I could imagine several fates: perhaps all the dust rains out before it reaches the mainland, perhaps it reaches the mainland and is washed out by orographically enhanced precip? It seems possible that factoring in wet deposition would change the ratios in favor of Asian sources. And it would matter in terms of the mass budgeting of P.

The supplemental material has an interesting discussion of the use of Hf isotopes as a way to constrain distance dust (the Zr hosted Hf drops out along the dust trajectory). The use of this argument is not introduced in the main body that I could see. Perhaps I missed it. If not then is it necessary in the supplement? Certainly the argument bolsters the Asian dust argument but not in a slam dunk way. I would suggest either removing it or providing a better introduction to that section so the reader can understand early why we have gone all esoteric (assuming of course that Sr and Nd are mainstream).

The latter part of the discussion tackles the biogeochemical implications particularly around land

use change, etc., but does not do too much with comparative analysis. This is probably ok. However there might be an opportunity make comparisons where they can be made (perhaps some of the work in the Rockies by Neff and others or the Volcanics in Northern Arizona or to Hawaii and Puerto Rico where P has been specifically linked to dust fluxes.

In Fig 1 why not use the symbol shape as well as the color to designate the locations along the transect?

Regardless of these comments I think the ms should be published and should be well received as an example of how to combine tools from different disciplines to enhance our understanding of how the Earth surface functions.

Reviewer #2 (Remarks to the Author)

This paper presents results from a dust collection study in the Sierra Nevada of California. Passive dust collectors were used to sample the eolian flux at four locations along an elevational transect within a Critical Zone Observatory. The Sr and Nd isotopic composition of the collected dust was compared with values from the literature for the two most likely source areas, central Asia and the Central Valley of California. The dust falls between these two end members so a mixing model was used to calculate the relative contributions of these two sources. The amount of phosphorus, a major limiting nutrient and terrestrial ecosystems, was also calculated in the dust. This revealed that the flux of eolian phosphorus to soils in the study area is greater than the amount of phosphorus delivered by weathering of the underlying bedrock and is also greater (or equal to) the amount removed from the system by erosion. This leads to the significant and novel conclusion that dust is more important than local bedrock erosion in delivering nutrients to mountain soils.

Two main strengths of the paper are the quantification of the relative contributions of the two dust source areas, and the approach to quantifying the flux of eolian P. The conclusion that dust delivers more plant available nutrients to the soil than weathering of the bedrock does is novel and will likely be of interest to other researchers. This broader aspect of the study could be enhanced by an expanded treatment of the "biochemical implications" summarized in the final section. The suggestion that the Sierra Nevada (to a degree) represent mountains worldwide, and therefore that dust may be more significant than rock weathering in other mountain ranges around the world is very intriguing and represents a major broader impact of the study. As written though this final point seems underdeveloped.

Overall the paper is generally clearly written, although in places it seems a bit cursory. I realize that a major challenge in writing papers this short is to balance the amount of information presented in the main text verses that save for the supplemental materials. My impression in reading this is that more of the information from the supplemental materials could be presented in the main text to make the paper more convincing and coherent -- however that may not be possible if the text is already bumping up against a character limit.

Some aspects of the experimental design could be presented more clearly. One big question is the apparent mismatch between the number of samples collected in the field (four collectors at monthly intervals) and the relatively small number of data points plotted in Figure 2. Were not all of the samples analyzed? If so, why and how were the analyzed sample selected? Or were they combined? If so, why? And why collect monthly if the samples were consolidated before analysis? Similarly, why are there unequal numbers of different colored symbols in Figure 2? In the Supplemental materials it sounds like three of the sites were sampled an equivalent number of times, with the fourth one (Shorthair) was sampled a smaller number. Why is this?

From a data presentation perspective, I think it would be helpful if a legend were included in Figures 2 and 3 to identify the study sites. I realize that for efficiency the authors expect the reader to refer back to Figure 1, but that is cumbersome and gets in the way of clearly

understanding the information presented in a later figures.

A few other points organized by line number:

Line 33 – why “relative importance”? It seems here you are trying to quantify and directly determine the importance of dust.

Line 34 – should also point out that there is much uncertainty about how the dust system will change in the future.

Line 51 – as noted above, it’s confusing that there so few data points shown in Figure 2 when four samplers were emptied monthly. This should be clarified.

Line 58 – it seems awkward to end with the footnote

Line 61 – inferred seems the wrong word, maybe “calculated”? Or “extrapolated”?

Line 72 – I see how I could work this out from comparing Figures 1 and 2, but you should be more explicit about how the trend is shown in Figure 2.

Line 82 – no dust from the Great Basin? I see in the supplemental information that you argue the height of the Sierra Nevada blocks dust from that direction, but is there any proof that this is the case? Some Hysplit data or atmospheric models?

Line 94 – is dust the only control on these measurements of poor air quality? It seems urbanization, transportation, refineries etc. could play a significant role.

Line 100 – I understand what you’re saying here, but isn’t it possible that comparing bulk rock chemistry of the granite to very fine-grained dust samples is a bit of apples and oranges? Would a better comparison be to extract the fine fraction derived from the granite and analyze that? Then again maybe with no sign of overlap between the dust and the bulk granite that wouldn’t be necessary...

Line 105 – should explain more how you define the end members since both the Central Valley and the Asian dust exist in fairly large ranges. Did you pick the mean values? Max/min? Some other approach?

Line 110 –I think it would be helpful to plot data versus elevation instead of expecting the reader to recall the relative order of the colors seen initially in Figure 1

Line 113 – doesn’t Figure 3 show days since last precipitation, rather than days during the sampling period? Or was all sampling conducted in one single rain free period?

Line 115 – once again I think including a legend for the symbology in Figure 3 would be very helpful, rather than requiring reference back to figure 1

Line 125 – all this makes sense, but why would these long-term changes in Asian wind strength and Central Valley soil moisture only influence the composition of the dust at the mid-elevation sites? Does that reveal something important about airflow patterns?

Jeff Munroe
Middlebury College

Reviewer #3 (Remarks to the Author)

The MS “Dust outpaces bedrock in nutrient supply to montane forest ecosystem” by Aciego, Riebe, Hart, Blakowski, Carey, Aarons, Dove, and Aronson, provides an assessment of dust input as important source of nutrients in montane forest ecosystems. The authors determine on a transect of four sites of different elevation the provenance and contribution of dust using Sr and Nd isotopes. Surely the theme is of broad interest for an international readership, and both methodology and science sound good. As far as I'm concerned, the geochemical data are mostly of high quality but seem to be not fully appreciated from the authors themselves. The MS is potentially good but it is often difficult to read, also because the data set is reported only in the Supplementary Online Material-Section. I believe that a table with the elemental and isotopic data and the results of modeling should find place in the MS for reader's convenience.

Additional weaknesses of the MS are as follows:

- 1) Nd isotopes proposed in the “INTRODUCTORY PARAGRAPH” as tracers for the Asian dust contribution are used in figure 2 only for providing evidence of nil or negligible dust contribution from the Sierra Nevada granite. However, the ϵ_{Nd} data of the Providence site fall out of the main array (fig.2) between the Central Valley and Asia dust end-members despite of the similar Sm/Nd ratios of others sites, and this is not explained in the text.
- 2) The contribution from the Asian dust has been evaluated by using Stewart's equation of isotopic mass balance of two end-members and in particular, only with Sr isotopes. Furthermore, in the excel data sheet, there are two Asian end-members (“generic “ Asian dust with $87Sr/86Sr = 0.714$ and Gobi dust with $87Sr/86Sr = 0.727$), but the minimum and maximum contribution from Asian end-member come only from the Gobi end-member. Why?.
- 3) The Hf isotopic data should be included in the main MS as additional prove of the Asian dust input.
- 4) The reference list in both The MS and Supplementary on line materials is careless, as is a very poor “exercise “ of copy and paste.
- 5) Figure 1 lacks of North indication as usually required for maps.
- 6) Figure 3 shows the variability of Sr isotopes relative to the days elapsed since the last rain precipitation, but the Authors do not discuss this figure in the text! Line 252: what means (<0.005)?
- 7) Figure 4 can be more convincing if coupled with a table.
- 8) Captures to the figures in the Supplementary Online Material-Section are absent.

Document Addressing Reviewer Comments for

Dust outpaces bedrock in nutrient supply to montane forest ecosystems

Aciego, S. M.^{1,2*}, Riebe, C. S.², Hart, S.C.³, Blakowski, M. A.¹, Carey, C.J.⁴, Aarons, S. M.¹, Dove, N.C.³, Botthoff, J.K.⁵, Sims, K.W.W.², and Aronson, E.⁴

Reviewer #1 (Remarks to the Author):

This manuscript makes use of Sr and Nd isotopes measured in dust collected from a west to east transect into the heart of the Sierra Nevada in California. The isotope data allow the authors to distinguish between local and distance sources of dust, which in this region they argue must be coming from Asia – they assume that the source is most likely the Gobi Desert. They find that between about 20 and 45% of the dust is derived from Asia and that the fraction is greater at the higher elevation sites (further away from local sources in the central valley). They then use the dust flux data in aggregate to calculate the amount of phosphorus (P) deposited on the forests at their higher elevation sites and to compare that with the amount of P delivered to the ecosystem by weathering of bedrock (here they assume that the erosion rate calculated using datasets that provide short-term and long-term erosion rates). They conclude that on sites with little soil and lots of bare rock most of the P comes from dust because of slow rates of soil production and that that fraction declines in locations where a thicker soil cover and hence faster soil production occurs. Finally they argue that given the increase in local dust associated with land-use intensification and potentially with increases in distance dust for the same reason, it is likely that these mountain forests and perhaps others like them elsewhere in the world are likely to see increased nutrient supplies in the future.

This is an excellent paper that combines contributions from isotope geochemists, geomorphologists and ecosystem scientists. The paper is well argued and well supported by the supplemental material. I have a few questions.

The crux of the isotope argument is that there are distinct isotopic signatures between the regional central valley sources and the long distance Asian sources. In the body of the text there is no discussion of other possible regional sources (I am thinking sources derived either from the northeast or the southeast of the mountains). The supplemental material contains a spirited defense of why one does not need to be concerned about those sources and earlier work on soils tends to support the idea that dust contributions from the basin and range drop off as one moves from the White Mountains or the Lahontan Basin into the Sierra Nevada. Still it might be good to bolster the argument with a sentence or two in the main body of the text since many will not interrogate the supplemental

material. **And for completeness it would be good to provide some sense of the isotope values for basin and range dust in the supplemental material.**

Unfortunately there is no published dust source (e.g., sediment) data for the basin and range of eastern California, northwestern Nevada, western Oregon or western Washington. However, we were able to make a comparison to the bedrock data available (Farmer & DePaolo, 1983); this has been added on Lines 168-175. We note that this is the same bedrock data used for basin and range comparison in Neff et al., 2008 on dust sources to the San Juan mountains in Colorado.

The dust data were collected during the summer of 2014. There is no discussion of what the winter delivery of mineral aerosol might be like. I could imagine that the delivery of local/regional dust could be suppressed by wet conditions but what is going to happen to the long-distance material? I could imagine several fates: perhaps all the dust rains out before it reaches the mainland, perhaps it reaches the mainland and is washed out by orographically enhanced precip? It seems possible that factoring in wet deposition would change the ratios in favor of Asian sources. And it would matter in terms of the mass budgeting of P.

We explicitly address this problem by comparing our data to the modeled annual dust flux data. See Lines 294-298

The supplemental material has an interesting discussion of the use of Hf isotopes as a way to constrain distance dust (the Zr hosted Hf drops out along the dust trajectory). The use of this argument is not introduced in the main body that I could see. Perhaps I missed it. If not then is it necessary in the supplement? Certainly the argument bolsters the Asian dust argument but not in a slam dunk way. **I would suggest either removing it or providing a better introduction to that section so the reader can understand early why we have gone all esoteric (assuming of course that Sr and Nd are mainstream).**

We have moved much of the supplementary data into the main text including the Hf data at the suggestion of this reviewer and reviewer 3.

The latter part of the discussion tackles the biogeochemical implications particularly around land use change, etc., but does not do too much with comparative analysis. This is probably ok. **However there might be an opportunity make comparisons where they can be made (perhaps some of the work in the Rockies by Neff and others or the Volcanics in Northern Arizona or to Hawaii and Puerto Rico where P has been specifically linked to dust fluxes.**

We have enlarged the biogeochemical implications sections to incorporate other

regions with granitic bedrock – e.g. Europe. See lines 384-396

One reason that we haven't explicitly compared our results to other studies of P in soils or lakes is that these prior studies have not measured the P flux directly from dust nor compare dust to regolith production. Because the results are not directly comparable, we believe that this would be a tangential discussion.

In Fig 1 why not use the symbol shape as well as the color to designate the locations along the transect?

We have modified the figures to be consistent across all for shape and color of symbols representing sampling sites, elevation and sampling date.

Reviewer #2 (Remarks to the Author):

This paper presents results from a dust collection study in the Sierra Nevada of California. Passive dust collectors were used to sample the eolian flux at four locations along an elevational transect within a Critical Zone Observatory. The Sr and Nd isotopic composition of the collected dust was compared with values from the literature for the two most likely source areas, central Asia and the Central Valley of California. The dust falls between these two end members so a mixing model was used to calculate the relative contributions of these two sources. The amount of phosphorus, a major limiting nutrient and terrestrial ecosystems, was also calculated in the dust. This revealed that the flux of eolian phosphorus to soils in the study area is greater than the amount of phosphorus delivered by weathering of the underlying bedrock and is also greater (or equal to) the amount removed from the system by erosion. This leads to the significant and novel conclusion that dust is more important than local bedrock erosion in delivering nutrients to mountain soils.

Two main strengths of the paper are the quantification of the relative contributions of the two dust source areas, and the approach to quantifying the flux of eolian P. The conclusion that dust delivers more plant available nutrients to the soil than weathering of the bedrock does is novel and will likely be of interest to other researchers. This broader aspect of the study could be enhanced by an expanded treatment of the “biochemical implications” summarized in the final section. The suggestion that the Sierra Nevada (to a degree) represent mountains worldwide, and therefore that dust may be more significant than rock weathering in other mountain ranges around the world is very intriguing and represents a major broader impact of the study. **As written though this final point seems underdeveloped.**

We have added additional text to the biogeochemical implications section to incorporate other regions. See lines 384-396

Overall the paper is generally clearly written, although in places it seems a bit cursory. I realize that a major challenge in writing papers this short is to balance the amount of information presented in the main text verses that save for the supplemental materials. **My impression in reading this is that more of the information from the supplemental materials could be presented in the main text to make the paper more convincing and coherent -- however that may not be possible if the text is already bumping up against a character limit.**

Much of the supplemental material has been moved to the main text.

Some aspects of the experimental design could be presented more clearly. **One big question is the apparent mismatch between the number of samples collected in the field (four collectors at monthly intervals) and the relatively**

small number of data points plotted in Figure 2. Were not all of the samples analyzed? If so, why and how were the analyzed sample selected? Or were they combined? If so, why? And why collect monthly if the samples were consolidated before analysis? Similarly, why are there unequal numbers of different colored symbols in Figure 2? In the Supplemental materials it sounds like three of the sites were sampled an equivalent number of times, with the fourth one (Shorthair) was sampled a smaller number. Why is this?

The sampling dates and number of samples have been clarified, see lines 415-425 and Table 1.

From a data presentation perspective, I think it would be helpful if a legend were included in Figures 2 and 3 to identify the study sites. I realize that for efficiency the authors expect the reader to refer back to Figure 1, but that is cumbersome and gets in the way of clearly understanding the information presented in a later figures.

There is not space to put a legend in all of the figures, so we instead described the shape and color scheme in each figure caption.

A few other points organized by line number:

Line 33 – why “relative importance”? It seems here you are trying to quantify and directly determine the importance of dust.

Modified as suggested, line 35

Line 34 – should also point out that there is much uncertainty about how the dust system will change in the future.

We agree with this point, and have added text in the section “biogeochemical implications” *lines 384-396*

Line 51 – as noted above, it’s confusing that there so few data points shown in Figure 2 when four samplers were emptied monthly. This should be clarified.

The sampling dates and number of samples have been clarified, see see lines 415-425 and Table 1.

Line 58 – it seems awkward to end with the footnote

We have modified citations within the text so that they are not referred to directly

but instead indicate the reference author and then footnote the author.

Line 61 – inferred seems the wrong word, maybe “calculated”? Or “extrapolated”?

Text has been modified to “extrapolated”, line 76

Line 72 – I see how I could work this out from comparing Figures 1 and 2, but you should be more explicit about how the trend is shown in Figure 2.

We added Figure 3c,d, which provide an explicit comparison of isotopic composition to elevation and changed the text to refer to Figure 3 instead. Lines 84-90

Line 82 – no dust from the Great Basin? I see in the supplemental information that you argue the height of the Sierra Nevada blocks dust from that direction, but is there any proof that this is the case? Some Hysplit data or atmospheric models?

We moved the discussion of airmasses to the main text (lines 114-115), and provided a seasonal Hysplit modeling record to the supplemental and bedrock data from the Great Basin (no dust data is published) to lines 168-175.

Line 94 – is dust the only control on these measurements of poor air quality? It seems urbanization, transportation, refineries etc. could play a significant role.

We have modified the sentence to say “contributed to the higher number of poor air quality days”, lines 143. The document cited actually states that “Unfortunately, many cities suffered more spikes in short-term particle pollution, particularly in the West, where continuing drought and heat may have increased the dust....”

Line 100 – I understand what you’re saying here, but isn’t it possible that comparing bulk rock chemistry of the granite to very fine-grained dust samples is a bit of apples and oranges? Would a better comparison be to extract the fine fraction derived from the granite and analyze that? Then again maybe with no sign of overlap between the dust and the bulk granite that wouldn’t be necessary...

Weathering (as described in the text, lines 109-111) should not change the isotopic composition of the Nd, so for the purposes of discounting the Sierra Nevada granite, comparing dust to granite is valid.

Line 105 – should explain more how you define the end members since both the Central Valley and the Asian dust exist in fairly large ranges. Did

you pick the mean values? Max/min? Some other approach?

We used the mean composition of the Central Valley and Asian dust as the compositions for the fractional compositions; see lines 195-197.

Line 110 –I think it would be helpful to plot data versus elevation instead of expecting the reader to recall the relative order of the colors seen initially in Figure 1

We have added two panels to Figure 3 to provide elevation context.

Line 113 – doesn't Figure 3 show days since last precipitation, rather than days during the sampling period? Or was all sampling conducted in one single rain free period?

All sampling was conducted in one single rain-free period (the continuing drought in 2014), line 68

Line 115 – once again I think including a legend for the symbology in Figure 3 would be very helpful, rather than requiring reference back to figure 1

We have added descriptions of the symbology to each of the figure captions.

Line 125 – all this makes sense, but why would these long-term changes in Asian wind strength and Central Valley soil moisture only influence the composition of the dust at the mid-elevation sites? Does that reveal something important about airflow patterns?

We have modified the text for clarification (lines 225-229); because we only have 1 data point for the highest elevation site, we cannot make a temporal inference, hence "mid-elevation sites".

Reviewer #3 (Remarks to the Author):

The MS “Dust outpaces bedrock in nutrient supply to montane forest ecosystem” by Aciego, Riebe, Hart, Blakowski, Carey, Aarons, Dove, and Aronson, provides an assessment of dust input as important source of nutrients in montane forest ecosystems. The authors determine on a transect of four sites of different elevation the provenance and contribution of dust using Sr and Nd isotopes. Surely the theme is of broad interest for an international readership, and both methodology and science sound good. As far as I'm concerned, the geochemical data are mostly of high quality but seem to be not fully appreciated from the authors themselves. The MS is potentially good but it is often difficult to read, also because the data set is reported only in the Supplementary Online Material-Section. **I believe that a table with the elemental and isotopic data and the results of modeling should find place in the MS for reader's convenience.**

The format of Nature Communications provides tables as links, so we actually believe that a supplementary data table in Excel format is actually more useable for any readers than a text formatted table.

Additional weaknesses of the MS are as follows:

1) Nd isotopes proposed in the “INTRODUCTORY PARAGRAPH” as tracers for the Asian dust contribution are used in figure 2 only for providing evidence of nil or negligible dust contribution from the Sierra Nevada granite. However, the ϵNd data of the Providence site fall out of the main array (fig.2) between the Central Valley and Asia dust end-members despite of the similar Sm/Nd ratios of others sites, and this is not explained in the text.

The Nd isotope values of all of the samples fall between -4 and -6, well within the range of the Central Valley (-4 to -6) and Asia (-5 to -9).

2) The contribution from the Asian dust has been evaluated by using Stewart's equation of isotopic mass balance of two end-members and in particular, only with Sr isotopes . Furthermore, in the excel data sheet, there are two Asian end-members (“generic “ Asian dust with $87\text{Sr}/86\text{Sr} = 0.714$ and Gobi dust with $87\text{Sr}/86\text{Sr} = 0.727$), but the minimum and maximum contribution from Asian end-member come only from the Gobi end-member. Why?

We apologize for the confusion in presenting Asia and Gobi as different endmembers. The two dominant sources of Asian dust that are transported across the Pacific are the Gobi Desert and Taklimakan deserts, but the work of Igarashi et al. 2011 suggests that the Gobi is the dominant source. We used the average composition of size-appropriate dust source material from the Gobi

(0.714) and Taklimakan (0.727) deserts to make our two calculations. We also updated Supplementary Table 1 to indicate that these are the two sources we are comparing. We have updated the text providing this information in lines 120-136 and 196-205

3) The Hf isotopic data should be included in the main MS as additional prove of the Asian dust input.

We have incorporated the supplementary material into the main text, including the Hf data.

4) The reference list in both The MS and Supplementary on line materials is careless, as is a very poor “exercise “ of copy and paste.

We have edited the references so that they follow the format required for Nature Communications.

5) Figure 1 lacks of North indication as usually required for maps.

A north arrow has been added.

6) Figure 3 shows the variability of Sr isotopes relative to the days elapsed since the last rain precipitation, but the Authors do not discuss this figure in the text! Line 252: what means (<0.005)?

We discuss figure 3 in lines 84-90 and 208-217. In the figure caption, we also updated the distribution in sample values so it is 2 SD instead of a range.

7) Figure 4 can be more convincing if coupled with a table.

The data for figure 4 is presented in Supplementary Table 2.

8) Captures to the figures in the Supplementary Online Material-Section are absent.

Captions have been added to the additional figures and the new supplementary figures.

Reviewers' Comments:

Reviewer #1 (Remarks to the Author)

I have reviewed the authors' response to reviewer comments and the resulting manuscript. I believe that the manuscript is now ready for publication.

Reviewer #2 (Remarks to the Author)

The paper was nicely improved by revision and I appreciate your careful attention to the comments I and the other reviewers provided. The paper presents a stronger, clearer, and more convincing argument now that much material was moved from the supplemental file and the figure captions were improved. I particularly appreciate the updates made emphasize how the Sierra Nevada are broadly representative of many mountain ranges worldwide, and the details added about possible dust sources to the east of these mountains. Congratulations on a successful and significant project.

Reviewer #3 (Remarks to the Author)

I believe that MS may be published in Nature .com